# Butyrate and Class I Histone Deacetylase Inhibitors Promote Differentiation of Neonatal Porcine Islet Cells into Beta Cells

**DOI:** 10.3390/cells10113249

**Published:** 2021-11-19

**Authors:** Yichen Zhang, Yutian Lei, Mohsen Honarpisheh, Elisabeth Kemter, Eckhard Wolf, Jochen Seissler

**Affiliations:** 1Medizinische Klinik und Poliklinik IV, Diabetes Zentrum-Campus Innenstadt, Ludwig-Maximilians-Universität Klinikum München, 80336 München, Germany; yichen.zhang@med.uni-muenchen.de (Y.Z.); yutian.lei@med.uni-muenchen.de (Y.L.); Mohsen.honarpisheh@med.uni-muenchen.de (M.H.); 2Institute of Molecular Animal Breeding and Biotechnology, Gene Centre and Department of Veterinary Sciences, Ludwig-Maximilians-Universität München, 85764 Oberschleissheim, Germany; kemter@genzentrum.lmu.de (E.K.); ewolf@genzentrum.lmu.de (E.W.); 3German Centre for Diabetes Research (DZD), 85764 Oberschleissheim, Germany

**Keywords:** islet development, short chain fatty acids, butyrate, HDAC inhibitor, porcine islets, NPICCs, beta cell differentiation

## Abstract

Neonatal porcine islets-like clusters (NPICCs) are a promising source for cell therapy of type 1 diabetes. Freshly isolated NPICCs are composed of progenitor cells and endocrine cells, which undergo a maturation process lasting several weeks until the normal beta cell function has developed. Here, we investigated the effects of short-chain fatty acids on the maturation of islet cells isolated from two to three day-old piglets. NPICCs were cultivated with acetate, butyrate and propionate (0–2000 µM) for one to eight days. Incubation with butyrate resulted in a significant upregulation of insulin gene expression and an increased beta cell number, whereas acetate or propionate had only marginal effects. Treatment with specific inhibitors of G-protein-coupled receptor GPR41 (β-hydroxybutyrate) and/or GPR43 (GPLG0974) did not abolish butyrate induced insulin expression. However, incubation of NPICCs with class I histone deacetylase inhibitors (HDACi) mocetinostat and MS275, but not selective class II HDACi (TMP269, MC1568) mimicked the butyrate effect on beta cell differentiation. Our study revealed that butyrate treatment has the capacity to increase the number of beta cells, which may be predominantly mediated through its HDAC inhibitory activity. Butyrate and specific class I HDAC inhibitors may represent beneficial supplements to promote differentiation of neonatal porcine islet cells towards beta cells for cell replacement therapies.

## 1. Introduction

Type 1 diabetes (T1D) is a chronic autoimmune disease leading to the complete destruction of insulin-producing beta cells. Clinical studies have shown that islet cell transplantation is an effective treatment option in some patients with T1D. However, the clinical application of pancreas and islet cells transplantation is limited by the shortage of organ donors and problems in the isolation of large numbers of high quality islets from organ donors [1].

Porcine islets are a promising source for islet cell replacement therapies due to the high similarity in the regulation of glucose homeostasis as compared to human islets, the unlimited availability and the novel molecular techniques to generate less immunogenic porcine tissues by genetic engineering of donor pigs [2,3,4]. Previous studies have demonstrated that pancreata from neonatal, pre-weaning piglets have several advantages over adult pigs, including an easier isolation process to yield high quality islets and lower costs [5,6,7]. Neonatal porcine islets-like clusters (NPICCs) are immature with respect to glucose-stimulated insulin secretion (GSIS), and display some characteristics of islet progenitor cells. After transplantation, NPICCs from young piglets need 4–12 weeks to restore normoglycemia and normal glucose tolerance in immunocompromised diabetic mice [8,9]. It is thought that this spontaneous maturation process includes differentiation of islet and ductal progenitor cells into endocrine cells as well as beta cell proliferation [10,11]. How to improve and accelerate this postnatal differentiation is still poorly understood.

Further improvements of culture conditions after NPICC isolation are necessary to get more insight into the regulatory mechanisms involved in beta cell maturation in order to optimize the cell product for NPICC transplantation. Positive effects on in vitro maturation have been described by treatment with glucagon-like peptide 1 (GLP-1) and cholecystokinin [12], exendin-4 [11,13], a combination of insulin-like-growth factor-1, foetal calf serum and sodium butyrate [14], and prolongation of the culture period up to 12 days [15]. Several studies have shown that some short-chain fatty acids (SCFAs, chain length of one to five carbon atoms) directly promote the survival and function of beta cells. It has been reported that acetate and propionate increased GSIS in mouse and human islets [16,17], whereas other studies found an inhibition of GSIS in mouse and rat islets by propionate [18,19,20]. Recently, no significant effects of acetate and propionate on insulin and glucagon secretion were detected using the perfused mouse pancreas model [21]. In a perfusion model, butyrate significantly suppressed insulin secretion at high but not at low glucose concentrations [21]. These conflicting results suggest that acetate, propionate and butyrate may have differential effects on different islet cell types. In addition, it is possible that interspecies differences exist between human and rodent islets. Metabolic studies using SCFAs and porcine islets are not available thus far.

In the present study, we investigated the potential of SCFAs to optimize NPICC culture conditions and promote differentiation of islets into insulin-positive porcine beta cells. We found that butyrate and specific inhibitors of histone deacetylases (HDAC) significantly upregulated insulin gene and protein expression, suggesting that butyrate treatment may be useful to enhance beta cell maturation and increase beta cell mass.

## 2. Materials and Methods

### 2.1. Isolation and Culture of NPICCs

All animal experiments were approved by the local animal welfare authority and were performed in agreement with Directive 2010/63/EU. NPICCs were isolated from the donor pancreas of two to three day–old piglets by collagenase digestion using a protocol developed by Korbutt et al. [8], which was modified as described previously [9]. Cells were cultured in 10-cm cell culture dishes (Corning, Wiesbaden, Germany) in basal islet culture (B-IC) medium, composed of RPMI 1640 medium (PAN-Biotech, Aidenbach, Germany), 2% human serum albumin (Takeda, Konstanz, Germany), 10 mM nicotinamide, 20 ng/mL exendin-4 (Merck, Darmstadt, Germany) and 1% antibiotic-antimycotic (Thermo Fisher Scientific, Germering, Germany) at 37 °C in a humidified atmosphere of 95% air and 5% CO_2_. Medium was changed at daily (days one to four) and then every second day. At culture day three, NPICCs were counted and expressed in islet equivalents (IEQ, 1 IEQ is equal to an islet with a diameter of 150 μm) using a stereomicroscope followed by incubation in B-IC medium with sodium acetate (0, 100, 500, 1000, 2000 µM), sodium butyrate (0, 100, 500, 1000, 2000 µM), or sodium propionate (0, 100, 500, 1000, 2000 µM) (Merck, Darmstadt, Germany) for an additional one, two, four, six, and eight days, respectively. Cell viability was determined by trypan blue exclusion test and propidium iodide/fluorescein diacetate dye staining (Thermo Fisher Scientific, Germering, Germany). Islet yield was expressed as the percentage of NPICCs compared to basal medium.

### 2.2. Glucose Stimulated Insulin Release Assay (GSIS)

NPICCs were harvested after incubation for six days in B-IC medium with and without 1000 µM butyrate and washed two times in a Krebs-Ringer buffer (KRB: 135 mM NaCl, 4.8 mM KCL, 1.2 M Mg_2_SO_4_, 1.2 mM KH_2_PO_4_, 1.3 mM CaCl_2_, 5 mM NaHCO_3_, 10 mM HEPES, 0.5% BSA pH 7.4). Then 100 IEQs were seeded in duplicates in 24-well plates in KRB buffer with low glucose (2.8 mM) for 1 h followed by a 1 h incubation in low glucose or high glucose (20.0 mM) at 37 °C, 5% CO_2_. The supernatant from each sample was collected and stored at −80 °C. Insulin concentration was measured by a porcine insulin ELISA (Mercordia, Uppsala, Sweden) according to the manufacturer’s instructions. Stimulation indices (SI) were calculated by dividing the insulin concentration secreted in high glucose by that secreted in low glucose.

### 2.3. Insulin and DNA Content

Duplicates of 100 islet equivalents (IEQ) were incubated with or without 1000 µM butyrate for six days, washed with PBS, lysed in 2 mM acetic acid buffer, 0.25% bovine serum albumin and sonicated with five pulses at 1J on ice for 30 s by using Branson SFX 150 Digitaler Sonifier (Branson, Dietzenbach, Germany). Then samples were centrifuged at 800× *g* for 15 min at 4 °C. Supernatants were collected and stored at −80 °C. Insulin concentration was measured by a porcine Insulin ELISA Kit (Mercodia, Uppsala, Sweden) according to the manufacturer’s instructions.

Other islet cell aliquots (100 IEQ) were resuspended in citrate buffer (150 mM NaCl, 15 mM citrate, 3 mM EDTA pH 7.4) and centrifuged at 200× *g* for 10 min at 4 °C. Cell pellets were resuspended in 10 mM Tris-buffer, 1 mM EDTA, pH 7.5 and DNA concentration was assayed by Quant-iT™ dsDNA Assay Kit (Thermo Fisher Scientific, Germering, Germany) following the manufacturer’s instructions. Insulin content was normalized to the sample DNA content and expressed as µU of insulin/ng of DNA.

### 2.4. Flow Cytometry Analysis

NPICCs were dissociated into single cells by treatment with TrypLE (Thermo Fisher Scientific, Germering, Germany) and washed with PBS + 10% foetal calf serum (FCS) and filtered through a 30-µm pre-separation filter (Miltenyi, Bergisch-Gladbach, Germany). Then cells were fixed/permeabilized using an intracellular staining buffer set and incubated with Fc-Block (anti-mouse CD16/CD32) for 10 min at room temperature (Thermo Fisher Scientific, Germering, Germany). Intracellular staining was performed using the following fluorochrome-labeled antibodies: mouse anti-insulin AF647 (clone T56-107), mouse anti-glucagon-PE (clone U16-850), mouse anti-somatostatin AF488 (clone U24-3545), mouse anti-Pdx-1-PE (clone 658A5), mouse anti-Pdx-1-AF488 (clone 658A5), mouse anti-Nkx6.1-AF647 and mouse anti Nkx6.1-PE (clone R11-560) (all from BD Biosciences, Heidelberg, Germany). All antibodies were pretested for appropriate dilution and for specificity using isotype control antibodies. Antibodies were incubated at 4 °C for 30 min followed by two washing steps with a permeabilization buffer. Flow cytometry data were acquired on a flow cytometer (BD Accuri™ C6 Plus Flow Cytometer, BD Biosciences, Heidelberg, Germany) and analyzed using FlowJo software version 10.4 (TreeStar, Ashland, USA).

### 2.5. Proliferation Assay

A Click-iT EdU Alexa Fluor 488 Imaging Kit (Thermo Fisher Scientific, Germering, Germany) was used to perform 5-Ethynyl-2′deoxyuridine (Edu) labeling. Briefly, NPICCs were cultured in B-IC medium in 6-well plates for six days in the presence of butyrate (1000 µM). EdU (10 µM) was added to the medium during the last 72 h. Then islets were embedded in Histogel and paraffin, sectioned and used for EdU staining according to the manufacturer’s instructions. Tissues were permeabilized with 0.5% Triton X-100 for 20 min and incubated with the Click-iT reaction cocktail for 30 min at room temperature. Slices were stained with polyclonal guinea pig anti-insulin antibody (diluted 1:300, Agilent-Dako, Frankfurt, Germany) for 1 h, and secondary antibody Alexa Fluor^®^ 594 AffiniPure donkey anti-guinea pig IgG (H+L) for 45 min (1:1000, Jackson ImmunoResearch, Cambridgeshire, UK). Images were taken and quantified by Leica DM2500 microscope (Leica, Wetzlar, Germany) and Image J software (ImageJ 1.52a, http://rsb.info.nih.gov/ij/download.html, accessed on 23 April 2018). At least 1000 cells per each NPICC preparations were analyzed to count Edu labelled cells (*n* = 6).

### 2.6. Studies on GPCR and HDAC Mediated Signaling

NPICCs were seeded in 24-well plates in B-IC medium in the presence or absence of 1000 µM butyrate plus/minus 5 mM β-hydroxybutyrate (BHB), a specific antagonist of GPR41, or 200 nM GPLG0974, a specific antagonist of GPP43, or a combination of both inhibitors (Merck, Darmstadt, Germany). After six days of incubation, cells were harvested and assessed for the expression of islet maturation and differentiation related cell markers by qRT-PCR. To study the effects of HDACs on NPICCs maturation we tested several specific small molecule HDAC is including class I selective inhibitor MS275 (1 µM) (inhibition of HDAC 1, 2, 3, and 8) and 1 µM mocetinostat (inhibition of HDAC 1, 2, 3, and 11) as well as class IIa selective inhibitors TMP269 and MC1568 (inhibition of HDAC4, 5, 7, and 9) (SelleckChem, Houston, USA). After six days, cells were harvested and processed for RT-qPCR.

### 2.7. Immunofluorescence

NPICCs were embedded in Histogel (Thermo Fisher Scientific, Germering, Germany) and fixed in 4% paraformaldehyde overnight. Four μm thick sections were permeabilized with 0.25% Triton X-100 in Tris-buffered saline (TBS), blocked with 5% donkey serum in TBS and incubated with polyclonal guinea pig anti-insulin (1:300, Agilent-Dako, Frankfurt, Germany) and rabbit anti-glucagon (1:200, Merck, Darmstadt, Germany) antibodies for 1 h at room temperature. Following washes with TBS, Alexa Fluor^®^ 594 AffiniPure donkey anti-guinea pig IgG (H+L) antibody and Alexa Fluor^®^ 488 AffiniPure donkey anti-rabbit IgG (H+L) antibody (1:1000, Jackson ImmunoResearch, Cambridgeshire, UK) were applied for 1 h at RT. For nuclear staining, DAPI (Vector Laboratories, Burlingame, CA, USA) was added. Images were taken and quantified by Leica DM2500 microscope and Image J software (ImageJ 1.52a, http://rsb.info.nih.gov/ij/download.html, accessed on 23 April 2018). The number of insulin-positive and glucagon-positive cells was determined and expressed as percentage of total cell number per islet.

### 2.8. HDAC Activity Assays

A fluorogenic HDAC assay kit (BPS Bioscience, San Diego, CA, USA) was used to assess the ability of butyrate to inhibit enzyme activity of recombinant class I HDAC according to the manufacturer’s instructions. Briefly, HDAC was incubated with butyrate (100, 500, 1000, 2000 µM) on 96-well plates for 1 h. HDAC substrate was added (5 μL of 1 mM DMSO stock solutions), and the plates were incubated at 37 °C for 30 min. Finally, developer solution was added for an additional 15 min at 37 °C.

To measure inhibition of HDAC activity by butyrate in NPICCs, an in situ HDAC activity fluorometric assay (BioVison, Milpitas, CA, USA) was used. One hundred NPICCs were seeded in 96-well plates and cultured in BI-C medium or medium supplemented with 1000 µM butyrate. After six days, the medium was removed and reaction mix (100 µL per well) was added to each well for 3 h at 37 °C. Then, 100 µL developer solution was added into each well for 30 min at 37 °C. Fluorescence signals were detected using a FLUOstar^®^ Omega Plate Reader (BMG LABTECH, Ortenberg, Germany), with excitation and emission filters of 355 nm and 460 nm. All the measures were performed in duplicates or triplicates.

### 2.9. RNA Extraction, Reverse Transcription and Quantitative Real-Time PCR

RNA was extracted using the ReliaPrep™ mRNA Cell and Tissue Miniprep System (Promega, Waldorf, Germany). Total RNA (500 ng) was reverse transcribed using random primers with and without GoScript™ Reverse Transcriptase (Promega, Waldorf, Germany). The qRT-PCR was carried out using 2 µL cDNA, 300 nM of forward and reverse primer and SsoFast™ EvaGreen^®^ Supermix (Bio-Rad, Feldkirchen, Germany) on MaxPro-Max3000P Real-time PCR system (Stratagene, La Jolla, CA, USA). Reaction conditions were as follows: 95 °C for 10 min, 40 cycles of 10 s at 95 °C and 20 s at 60 °C, and followed by a melting curve stage of 95 °C for 10 s and 60 °C for 20 s. The relative gene expression of insulin (*INS*), glucagon (*GCG*), pancreatic and duodenal homeobox 1 (*PDX1*)*,* Neurogenin 3 (*NGN3*)*,* NK6 Homeobox 1 (*NKX6.1*) and aldolase fructose-bisphosphate B (*ALDOB*) were measured by reverse transcription quantitative polymerase chain reaction (RT-qPCR) normalized against the expression level of the glyceraldehyde 3-phosphate dehydrogenase (*GAPDH*) gene via threshold cycle (Ct) value based on the comparative 2^−ΔΔCt^ method. No template was used as reverse transcription (RT) control. Results were expressed as a fold change compared to NPICCs cultured under basal conditions.

The mRNA expression of *GPR43* and *GPR41* was analyzed by PCR using T100 Thermal Cycler (Bio-rad, Feldkirchen, Germany) with the following program: 95 °C for 3 min, 45 cycles of 15 s at 95 °C and 20 s at 60 °C, 1 min at 72 °C, and finally 5 min at 72 °C. The PCR products were evaluated by 1.5% agarose gel electrophoresis in Tris-borate-EDTA buffer stained with ethidium bromide (Promega, Waldorf, Germany). Images were taken by Gel Doc Imaging System (Intas Science imaging, Göttingen, Germany). Amplification without reverse transcriptase (−) RT served as control. Primer sequences are listed in Table 1.

### 2.10. Statistics

Data are expressed as mean ± standard deviation (SD) for normal distribution or as median for non-normal distribution. The normality of distribution was checked by Shapiro-Wilk’s test. For normally distributed data, we used two-tailed Student’s *t*-tests for two groups and one-way ANOVA with Tukey post-test for multiple comparisons. For non-normally distributed data, we used Mann-Whitney U-tests for two groups and Kruskal–Wallis test with Dunn’s test for multiple comparisons, respectively. *p* value < 0.05 was considered significant. Statistical analysis was performed using GraphPad Prism 8.0 software (GraphPad, San Diego, CA, USA).

## 3. Results

### 3.1. Butyrate Strongly Increases Insulin Gene Expression in a Dose- and Time-Dependent Manner

To investigate the potential role of SCFAs to support beta-cell maturation, we cultured NPICCs from day three after isolation for one to eight days in our regular culture medium (B-IC medium), or in medium supplemented with 100–2000 µM acetate, butyrate and propionate, respectively. The NPICC count, determined as IEQ per g of pancreas (IEQ/g), was 15,880 ± 3315 (*n* = 9) at day three. Treatment with SCFAs did not affect NPICC yield compared to cells cultured in basal medium (day six: acetate 94.5 ± 7.6%, butyrate 94.8 ± 4.8%, propionate 87.7 ± 6.3% of controls). SCFA treatment did not affect cell viability measured by a live/dead assay. As illustrated in Figure 1A,D, there was a significant dose- and time-dependent up-regulation of *INS* gene expression after incubation with 500–2000 µM butyrate for six days (2.6–4.7-fold) and at 1000 µM butyrate for six to eight days (2.2–4.1-fold). Butyrate treatment also resulted in an increase of *GCG* (1.5–2.2-fold) and *PDX1* gene expression (1.8–3.2-fold). This was only significant at a butyrate concentration of 1000 µM for glucagon and at 500 and 1000 µM for Pdx1 for six days. In contrast, exposure to acetate had only a modest effect with an upregulation of insulin gene expression at 1000 µM for two days and propionate did not significantly affect mRNA expression of *INS*, *GCG* and *PDX1* (Appendix A). Based on the results, butyrate at the concentration of 1000 µM was selected to examine mRNA expression of the endocrine progenitor cell marker NGN3, the beta cell maturation marker *NKX6.1* and *ALDOB*, a marker of functionally immature beta cells. During butyrate exposure, mRNA expression of *NGN3* and *ALDOB* were decreased and *NKX6.1* gene expression was significantly upregulated, confirming the assumption that butyrate favors maturation of NPICCs toward a beta cell phenotype.

### 3.2. Butyrate Enhances Protein Expression of Beta Cell Maturation Markers

To further explore the effects of butyrate, NPICCs were analyzed by FACS before butyrate treatment and after exposure to 1000 µM butyrate for six days. As illustrated in Figure 2A–C, butyrate treatment significantly increased the number (43.8 ± 4.7% vs. 33.1 ± 3.0%) and the median fluorescence intensity of insulin positive cells compared to cells cultured in B-IC medium (*p* < 0.01). Incubation with butyrate also resulted in a higher proportion of mature beta cells, defined by insulin+ and Nkx6.1+ co-staining (33.4 ± 8.3% vs. 21.0 ± 8.0%). This effect was not mediated by increased cell proliferation, since analysis by immunohistochemistry did not detect differences in Edu-positive cells in the presence or absence of butyrate (Appendix A). Interestingly, the percentage of glucagon positive cells slightly decreased in butyrate treated cells (12.8 ± 5.0% vs. 18.6 ± 7.0%), whereas the number of somatostatin positive cells (10.1 ± 1.0% vs. 10.0 ± 2.7%) and Pdx1 positive cells (39.3 ± 6.1% vs. 43.2 ± 9.1%) was not altered.

Although there was a significant upregulation of insulin gene and protein expression, insulin content (51.7 vs. 29.6 µU/ng DNA) and GSIS was only moderately increased from 1.7 to 2.1 (Figure 2F) suggesting that NPICCs still have not acquired a physiological glucose response.

### 3.3. Butyrate Up-Regulated Insulin Expression Is Independent of Binding to G-Protein Coupled Receptors

To further explore potential mechanisms via which butyrate induced insulin gene and protein expression, we analyzed involvement of GPCRs, which are well-known targets of SCFAs. First, we demonstrated that GPR41 and GPR43 are expressed in NPICCs (Figure 3A). Treatment with butyrate and specific blockers of GPR41 (β-hydroxybutyrate) or GPR43 (GPLG0974) did not significantly reduce the upregulation of insulin expression (Figure 3B). These data suggest that the butyrate effect on NPICCs is independent of the activation of GPCR.

### 3.4. Butyrate Stimulates Insulin Transcription via Inhibition of HDAC

Since butyrate is a known class I and II HDAC inhibitor (HDACi), we tested the HDAC activity of butyrate-treated NPICCs in fluorometric enzyme assays. Butyrate and trichostatin A (TSA) inhibited class I HDAC activity in a cell-free assay and decreased HDAC enzyme activity in NPICCs (Figure 4A,B). To further elucidate which HDAC enzyme may induce a proendocrine effect, we tested several specific small molecule HDACis including class I inhibitor MS275 and mocetinostat as well as class IIa inhibitors MC1568 and TMP269.

Exposure of NPICCs to MS275 and mocetinostat strongly increased the expression of insulin, recapitulating the effect of butyrate in NPICCs (Figure 4B). Treatment with MC1568 and TMP269 had no significant effect. Peak insulin gene expression was observed after treatment with mocetinostat (3.1-fold increase) providing evidence that inhibition of class I HDACs enhanced differentiation of neonatal porcine islets. These findings suggest that butyrate may induce maturation of neonatal porcine islets towards a beta cell phenotype through its HDAC inhibitory activity.

## 4. Discussion

Transplantation of neonatal porcine islets-like clusters (NPICCs) has the potential to overcome the problem of a shortage of organ donors. Optimization of pretransplant islet culture conditions is of great importance to increase the number of high quality, functionally active islets. In the present study, we investigated the potential of SCFAs to stimulate NPICCs development and the maturation of pancreatic beta cells. We demonstrated that butyrate and class I specific HDACi strongly induced insulin expression in NPICCs, which provides novel options to improve cell replacement therapies.

Because several studies have shown that SCFAs may directly promote survival and function of human and rodent pancreatic beta cells [16,22,23,24], we tested whether acetate, butyrate and propionate can accelerate differentiation and maturation of cells residing in immature NPICCs from 2–3 days old piglets. Consistent with previous reports, we observed a spontaneous increase of the percentage of alpha and beta cells from day three to day nine cultures [8,11,25,26]. We here report that butyrate but not acetate or propionate has the potential to activate a maturation program and significantly increase gene and protein expression of beta cell maturation markers compared to our standard culture protocol. After butyrate treatment, NPICCs consist of about 65% hormone positive islet cells including 43% insulin-, 12% glucagon-, and 10% somatostatin-positive cells. The observed increase in the number of beta cells could be explained by replication of pre-existing beta cells and/or the differentiation of residing islet precursor cells [10,26]. Since we did not detect an increased proliferation rate, it is likely that butyrate mainly induced development of new beta cells from ductal or progenitor-like cells. Lineage tracing studies of ductal and progenitor cells are necessary to clarify this question.

Earlier reports described a two to threefold increase of insulin expression with 23% insulin positive cells in porcine fetal cell clusters cultured for one week in medium supplemented with 10% human serum with either nicotinamide or butyrate [25]. Lopez-Avalos described a three to fivefold increased insulin content and threefold increased insulin gene expression in hydrogel encapsulated NPICCs after 14 days incubation in medium with 10% FCS stimulated with 500 µM butyrate and 10 nM IGF-1 plus/minus nicotinamide [14]. In our NPICC isolation protocol we exclude serum supplement, because serum growth factors are not well defined, and it leads to an attachment of NPICCs on the plastic surface of the culture dishes followed by a spreading of attached cells and rapid loss of expression of endocrine and beta cell markers (data not shown). A similar high number of endocrine cells was reported in a recent study, which shows that a 20-days culture period of NPICCs in medium with sequential addition of dexamethasone and oncostatin M, nicotinamide/exendin-4, and TGF-beta1/thrombin generated 42% insulin and 17% glucagon positive cells [11].

Butyrate is a physiological stimulus of GPCR, including GPR41 and GPR43. Using specific GPR41 and GPR43 inhibitors, we demonstrated that the increase of insulin expression may not be mediated through binding to one of these receptors. Since butyrate triggered beta cell development, it was surprising that GSIS was only modestly increased. Although we observed an upregulation of maturation markers (NKX6.1) and a downregulation of markers of immature beta cells (ALDOB), this may be explained by the fact that the beta cells are still not completely matured. Another explanation is that the binding of butyrate to GPR41 and GPR43 may influence GSIS differentially. GPR41 is coupled to Gi/o, which decreases intracellular cAMP levels by inhibition of adenylyl cyclase and thereby inhibiting PKA and EPAC mediated insulin secretion. GPR43 stimulation is linked with both Gi/o and Gq, decreasing cAMP levels and enhancing GSIS by increasing cytoplasmic calcium concentrations and activating PKA [20,27,28]. In primary isolated human islets, butyrate did not significantly stimulate GSIS. Insulin secretion of human islets were inhibited when treated with specific GPR43 agonists [29]. Therefore, it can be speculated that butyrate treatment increased beta cell maturation, but in parallel attenuated GSIS of NPICCs by binding to GPR43. Further experimental studies with specific blockers of the Gαi and Gαq pathways are needed to determine how butyrate modulate regulation of insulin secretion in NPICCs.

Another part of the physiological effects of butyrate is the modulation of chromatin acetylation and gene transcription by inhibition of class I and II histone deacetylases (HDAC), which regulate gene transcription by deacetylation of histones and transcription factors [30,31]. In general, hyperacetylation is associated with increased gene expression while hypoacetylation favours gene silencing. Class I and class II HDAC are expressed in developing pancreas and in human and rat islets and are differentially regulated by metabolic and inflammatory mediators [32,33]. It has been shown that acetylation of histone 4 is involved in glucose-dependent activation of the insulin gene promotor in the mouse beta cell line MIN6 and that PDX1 interacts with HDAC1 and HDAC2 to downregulate insulin expression at low glucose levels [34,35]. In vivo studies reported on decreased beta cell apoptosis and increased beta cell proliferation through modification of histone acetylation in streptozotocin-diabetic mice after butyrate treatment [36]. We confirmed that butyrate inhibited histone deacetylase enzyme activity in NPICCs by using HDAC assays. Our study revealed that inhibition of class I HDAC, which includes HDAC1, HDAC2, HDAC3 and HDAC8, by specific small molecules such as mocetinostat strongly induced insulin expression but did not promote differentiation of alpha and delta cells in NPICCs and thereby mimicked the maturation effect induced by butyrate. Because class II HDACi MC1568 and TMP269 did not influence insulin expression our data suggest that class I HDACi specifically modify gene regions involved in porcine beta cell maturation. These findings are in contrast to previous studies in fetal rat pancreatic cells reporting on an upregulation of *NGN3* gene expression and a strong increase of alpha cells and pancreatic polypeptide (PP) expressing cells by treatment with butyrate, TSA and MS275, respectively [32]. Class I specific HDACi, MS275 and valproic acid strongly decreased insulin and somatostatin expressing cells and treatment with the selective class II HDACi, MC1568, enhanced rat beta and delta cells [37]. In agreement with our results, exposure of fetal rat explants to TSA and butyrate strongly enhanced differentiation of insulin-expressing beta cells [32]. These data suggests that there may be major differences in the regulation of proendocrine transcription factors and insulin gene expression between rat and porcine pancreatic cells. In NPICCs, class I HDACi may play the crucial role in the regulation of pancreatic endocrine cell differentiation.

There are some limitations to our study. We have not determined the molecular mechanisms how HDAC inhibition promotes beta cell differentiation and maturation. Butyrate effects were only evaluated in vitro and not in vivo. Further studies are needed to explore whether treatment with butyrate or HDACi provide NPICCs with an improved capacity to reverse diabetes after transplantation.

## 5. Conclusions

We have demonstrated here that butyrate is a strong beta cell differentiation and maturation trigger in NPICCs. Our study highlights the contribution of class I HDACi in the development of the cell fate of endocrine cells derived from neonatal porcine islets. The modifying of class I HDAC activity may be the major mechanism whereby butyrate favors beta cell maturation. These findings have great promise with regard to the development of novel, efficient and stable protocols for in vitro generation of beta cells from immature progenitor cells and for the development of beta cell replacement therapies.

## Figures and Tables

**Figure 1 cells-10-03249-f001:**
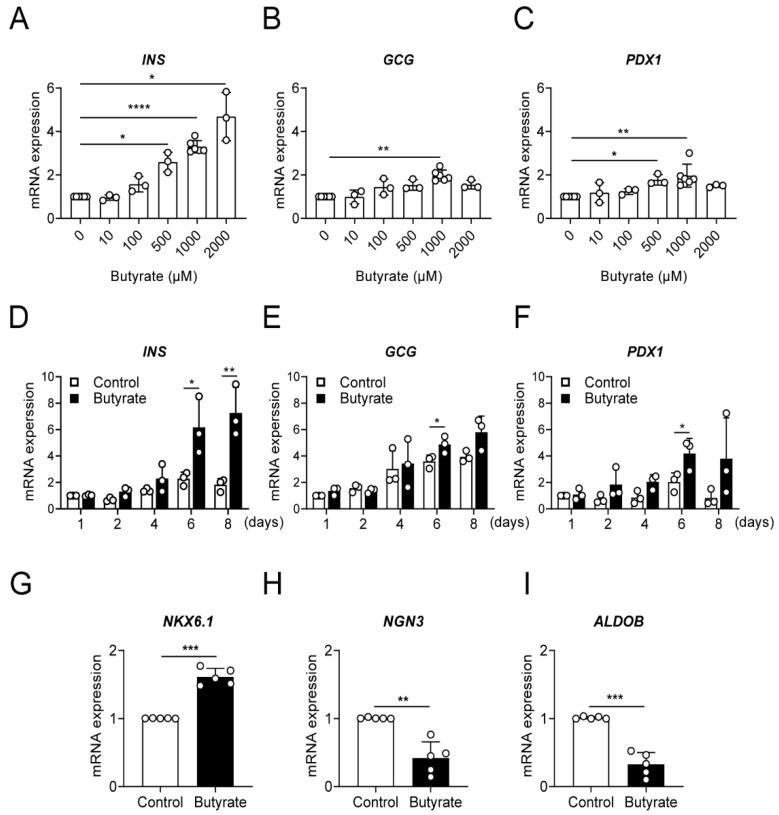
Exposure to butyrate significantly promotes beta cell maturation in a- and time-dependent manner. Relative gene expression of insulin (*INS*) (**A**,**D**), glucagon (*GCG*) (**B**,**E**), pancreatic and duodenal homeobox 1 (*PDX1*) (**C**,**F**), NK6 homeobox 1 *(**NKX6.1*) (**G**), neurogenin 3 (*NGN3*) (**H**), and aldolase fructose-bisphosphate B (*ALDOB*) (**I**) assessed by RT-qPCR in NPICCs. (**A**–**C**) NPICCs were treated with 100–2000 µM butyrate for six days, (**D**–**F**) 1000 µM butyrate for one to eight days and (**G**–**I**) 1000 µM butyrate for six days. Cells incubated without butyrate were used as controls. Data are presented as mean ± SD of three to six independent experiments. * *p*  <  0.05, ** *p*  <  0.01, *** *p*  <  0.001 and **** *p*  <  0.0001 vs. control groups.

**Figure 2 cells-10-03249-f002:**
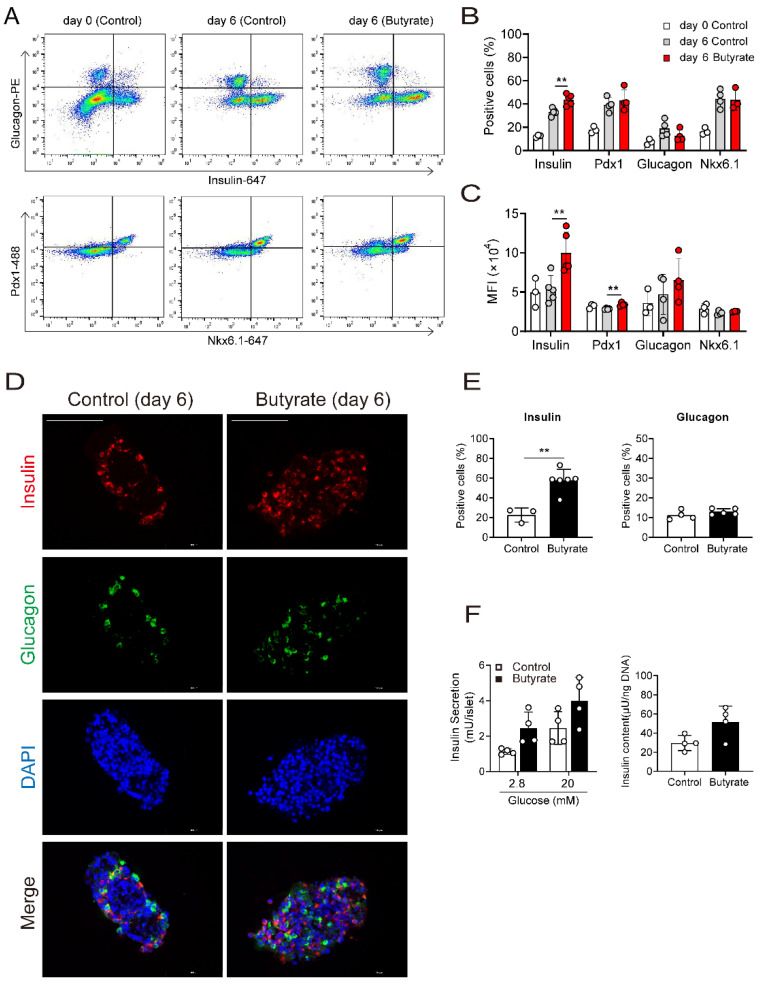
Butyrate increased the number of insulin positive cells and insulin content. (**A**) FACS analysis of NPICCs before and after treatment with butyrate (1000 µM) for six days. NPICCs were dispersed into single cells and analyzed by intracytoplasmic staining for insulin (AF647), glucagon (PE), Pdx1 (AF488) and Nkx6.1 (AF647). (**B**,**C**) Butyrate increased the number of insulin positive cells and the median fluorescence intensity (MFI) of insulin stained cells. (**D**,**E**) Immunofluorescence staining for insulin (red) and glucagon (green) confirmed increase of insulin positive NPICCs in the butyrate treated group (1000 µM for six days). scale bars = 100 μm. (**F**) GSIS and Insulin content were moderately increased in NPICCs treated with butyrate. Data are presented as mean ± SD (**B**,**E**,**F**) or median ± SD (**C**) of three to six independent experiments. ** *p*  <  0.01 vs. control group.

**Figure 3 cells-10-03249-f003:**
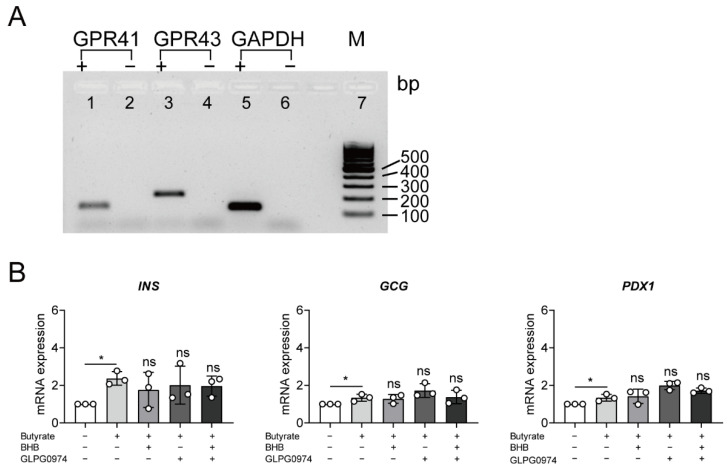
GPR41 and GPR43 are not involved in up-regulation of insulin gene expression by butyrate. (**A**) Expression of *GPR41* (119 bp) (lanes 1, 2), *GPR43* (197 bp) (lanes 3, 4) and *GAPDH* (lanes 5, 6) in NPICCs assessed by RT-PCR and agarose gel electrophoresis. Samples without reverse transcriptase (−) were included as controls (**B**) NPICCs were cultured in B-IC medium with 1000 µM butyrate for six days containing GPR41 inhibitor (5 mM β-hydroxybutyrate, BHB), or GPR43 inhibitor (200 nM GPLG0974) or both as indicated. The mRNA levels of *INS, GCG* and *PDX1* were examined by RT-qPCR. Cells without butyrate exposure were used as controls. Data are presented as mean ± SD of three independent experiments. * *p*  <  0.05 vs. control group. ns = not significant.

**Figure 4 cells-10-03249-f004:**
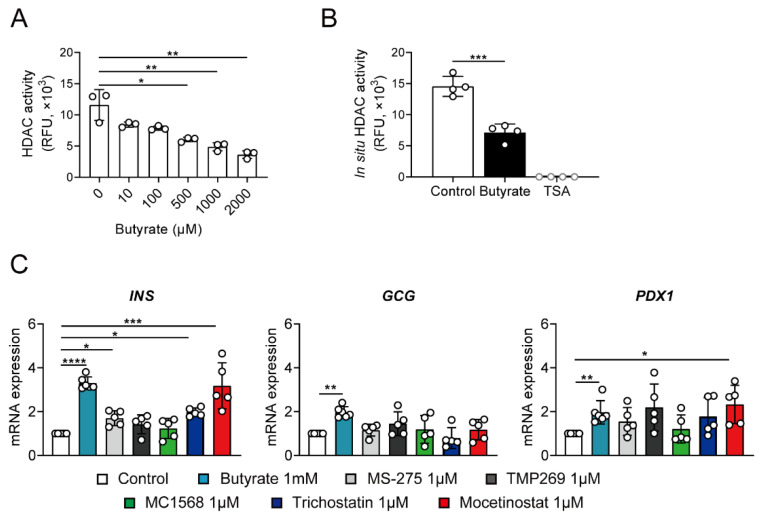
Inhibition of class I HDACs is involved in up-regulation of insulin gene expression. (**A**) Butyrate dose-dependently reduces in vitro HDAC enzyme activity in a HDAC fluorometric assay. (**B**) NPICCs were treated with 1000 µM butyrate and HDAC activity was measured by HDAC Activity Fluorometric Assay. Butyrate (1000 µM) and trichostatin A (TSA, 1 µM) significantly inhibited HDAC enzyme activity. (**C**) NPICCs were treated with different HDAC inhibitors for 6 days. TSA (1 µM) (class I and class II inhibitor), mocetinostat (1 µM) and MS275 (1 µM) (selective class I inhibitors) but not TMP269 (1 μM) or MC1568 (1 μM) (selective class II inhibitors) significantly increased insulin gene expression. NPICCs cultured in B-IC medium were used as controls. Data from three to five independent experiments are presented as mean ± SD. * *p*  <  0.05, ** *p*  <  0.01, *** *p* < 0.001, **** *p* < 0.0001 vs. control groups.

**Table 1 cells-10-03249-t001:** Primer sequences for quantitative real time-PCR.

Gene	Primer Sequence 5′-3′	Accession No.
*INS*	Forward: CAGGCCTTCGTGAACCAG	NM_001109772.1
	Reverse: CTTGGGCGTGTAGAAGAAGC	
*GCG*	Forward: GAATTCATTGCTTGGCTGGT	NM_214324.1
	Reverse: CATCTGAGAAGGAGCCATCAG	
*PDX1*	Forward: GTGGAAAAAGGAGGAGGACA	NM_001141984.3
	Reverse: CAGCTCCTCTCCCGAGGT	
*NKX6.1*	Forward: GCCTACCCCGTTTCAGTAGC	XM_021101796.1
	Reverse: GGGTGGACTCTGCATCACTC	
*NGN3*	Forward: GCCTGCGTCTCAGCTGAACTT	XM_021072424.1
	Reverse: AGCCAGAGGCAGGAGGAACAA	
*ALDOB*	Forward: ATTTGGAGGGCACTCTGTTG	XM_021066854.1
	Reverse: AGGTTGATAGCATTGAGGTTGAG	
*GPR43*	Forward: TCATGGGTTTCGGCTTCTACAG	EU122439.1
	Reverse: GTACTGAACGATGAACACGACG	
*GPR41*	Forward: ACTACTTCTCATCCTCGGGGTT	JX566879.1
	Reverse: CTCCACTTCGCTCTTCTTCAGT	
*GAPDH*	Forward: GTCGGTTGTGGATCTGACCT	NM_001206359.1
	Reverse: GTCCTCAGTGTAGCCCAGGA	

INS: insulin; GCG: glucagon; PDX1: pancreatic and duodenal homeobox 1; NGN3: neurogenin 3; NKX6.1: NK6 Homeo-box 1; ALDOB: aldolase fructose-bisphosphate B; GAPDH: glyceraldehyde 3-phosphate dehydrogenase; GPR43: G-Protein Coupled receptor 43; GPR41: G-Protein Coupled receptor 41.

## Data Availability

Data is contained within the article or Appendix A.

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
