# Peer review of "Butyrate and Class I Histone Deacetylase Inhibitors Promote Differentiation of Neonatal Porcine Islet Cells into Beta Cells"

_cells, 2021, doi:10.3390/cells10113249_

Round 1
Reviewer 1 Report
Comments:
- In the abstract section, the statement “Freshly isolated NPICCs are composed of progenitor cells and beta cells, which undergo a maturation process lasting several weeks until the normal beta cell function has developed” is not entirely correct. NPICCs are also composed of other endocrine cells such as alpha cells, delta cells and pancreatic polypeptide secreting cells (gamma cells).
- In the abstract, it was stated that islet cells were isolated from 2-3 days-old piglets. However, in the main document (Materials and Methods; Discussion) it was indicated that the islets were obtained from 2-5 days-old neonatal pigs.
- Introduction section, the authors indicated that type 1 diabetes can be cured by transplantation of human islets. Islet transplantation does not cure type 1 diabetes but it is a form of treatment for type 1 diabetes.
- The authors should add reference(s) to the statement “After transplantation, NPICCs need 4-12 weeks to restore normoglycemia and normal glucose tolerance in diabetic mice.” Is this statement true for various ages of neonatal pigs? Is this statement also true for both diabetic immune-deficient and immune-competent mice? It would be more appropriate to indicate whether this is based on the authors experience or based on the published literature.
- In the Methods, were the NPICCs cultured in 95% oxygen or 95% air?
- TrypE should be TrypLE?
- How many NPICCs were used for each assay described in section 2.2, 2.4, 2.5 (replace anti-duinea to anti-guinea), and 2.6 (change GRP43 to GPR43)
- In Flow cytometry methods, anti-CK7 antibody was mentioned but no results were shown using this particular antibody
- Figure 1 H and I are missing labels
- The authors mentioned that butyrate treatment significantly increased the number of and median fluorescence intensity of insulin positive cells resulting in a higher proportion of mature beta cells, defined by insulin+ and NKx6.1+ co-staining compared to cells cultured in B-IC medium (Fig 2B and 2C). This statement is not entirely correct as there is no significant difference between the percentages of NK6.1+ cells between butyrate treated islets and untreated control. As shown in Figure A, NPICCs were not stained for both insulin and NK6.1 but stained for glucagon and insulin or PDX-1 and NKx6.1. The antibodies used in the study for insulin and NK6.1 are both conjugated with AF647 so co-staining for both insulin and NKx6.1 is not feasible.
- Lines 295 and 371, change GRP41 to GPR41
- Figure 3 legend, add BHB = beta hydroxybutyrate
- Line 358, delete serum
- Line 386, change GRP43 to GPR43
- The authors should include in the discussion the limitation of their study such as the demonstration of butyrate treated NPICCs in reversing diabetes post-transplantation compared to untreated NPICCs.
Author Response
We thank the reviewers for taking the time to review our manuscript.
Response to Reviewer 1 Comments
Comments:
- In the abstract section, the statement “Freshly isolated NPICCs are composed of progenitor cells and beta cells, which undergo a maturation process lasting several weeks until the normal beta cell function has developed” is not entirely correct. NPICCs are also composed of other endocrine cells such as alpha cells, delta cells and pancreatic polypeptide secreting cells (gamma cells).
Response 1: We are grateful to the reviewer for pointing out this problem. We have re-written this part according to the Reviewer’s suggestion. Freshly isolated NPICCs are mainly composed of progenitor cells and endocrine cells….. (line 14)
- In the abstract, it was stated that islet cells were isolated from 2-3 days-old piglets. However, in the main document (Materials and Methods; Discussion) it was indicated that the islets were obtained from 2-5 days-old neonatal pigs.
Response 2: We are very sorry for this error. NPICC were isolated from 2-3 days-old piglets.
We changed 2-5 days to 2-3 days (line 79, line 392).
3 .Introduction section, the authors indicated that type 1 diabetes can be cured by transplantation of human islets. Islet transplantation does not cure type 1 diabetes but it is a form of treatment for type 1 diabetes.
Response 3: This sentence has been revised according to the comment of the Reviewer.
Clinical studies have shown that islet cell transplantation is an effective treatment option in some patients with Type 1 diabetes (lines 33-35).
- The authors should add reference(s) to the statement “After transplantation, NPICCs need 4-12 weeks to restore normoglycemia and normal glucose tolerance in diabetic mice.” Is this statement true for various ages of neonatal pigs? Is this statement also true for both diabetic immune-deficient and immune-competent mice? It would be more appropriate to indicate whether this is based on the authors experience or based on the published literature.
Response 4: We are grateful for these comments. To be more clear and in accordance with the reviewer concerns, we stated that this maturation phase is needed for neonatal islets after transplantation in SCID-mice and we have added some references.
NPICCs from young piglets need 4-12 weeks to restore normoglycemia and normal glucose tolerance in immunocompromised diabetic mice (Korbutt et al 1996, Wolf-van Buerck et al 2017) (lines 46-47)
- In the Methods, were the NPICCs cultured in 95% oxygen or 95% air?
Response 5: NPICCs were cultured in a humidified atmosphere of 95% air and 5% CO2
(line 85)
- TrypE should be TrypLE?
Response 6: This has been corrected in the revised manuscript. Thank you. (line 130)
- How many NPICCs were used for each assay described in section 2.2, 2.4, 2.5 (replace anti-duinea to anti-guinea), and 2.6 (change GRP43 to GPR43)
Response 7: For GSIS experiment and Insulin / DNA content measurement, 100 islet equivalents (IEQ) were used for each experiment, which means 100 - 120 islets. Because isolated islets and NPICCs differ in size (50-250 µm) IEQs are used to normalize the cell number. One IEQ is equal to an islet with a diameter of 150 μm. This definition is now described (line 87).
We apologize for the writing errors. Anti-duinea has been corrected to anti-guinea (line 159)
and GRP43 has been corrected to GPR43 in the revised manuscript (line 169)
- In Flow cytometry methods, anti-CK7 antibody was mentioned but no results were shown using this particular antibody
Response 8: CK7 results were not included. Therefore, we have deleted this sentence (lines 140-141)
- Figure 1 H and I are missing labels
Response 9: We have added the labels H and I to Figure 1. (page 7)
10.The authors mentioned that butyrate treatment significantly increased the number of and median fluorescence intensity of insulin positive cells resulting in a higher proportion of mature beta cells, defined by insulin+ and NKx6.1+ co-staining compared to cells cultured in B-IC medium (Fig 2B and 2C). This statement is not entirely correct as there is no significant difference between the percentages of NK6.1+ cells between butyrate treated islets and untreated control. As shown in Figure A, NPICCs were not stained for both insulin and NK6.1 but stained for glucagon and insulin or PDX-1 and NKx6.1. The antibodies used in the study for insulin and NK6.1 are both conjugated with AF647 so co-staining for both insulin and NKx6.1 is not feasible.
Response 10: We are sorry for the unclear description. In Figure 2 data are given for insulin+ and glucagon+ as well as Pdx-1+ and Nkx6.1+ staining. In addition, we performed a co-staining using insulin-AF647 and Nkx6.1-PE antibodies to determine the percentage of mature insulin+ and Nkx6.1+ cells. This has been corrected and is now described in more detail (lines 301-304)
- Lines 295 and 371, change GRP41 to GPR41
Response 11: Thank you. GRP41 has been corrected to GPR41 in the revised manuscript (line 334, line 418)
- Figure 3 legend, add BHB = beta hydroxybutyrate
Response 12: BHB means β-hydroxybutyrate. This is now added (line 346)
- Line 358, delete serum
Response 13: The extra word has been deleted (line 405)
- Line 386, change GRP43 to GPR43
Response 14: This is now changed in the revised manuscript. Thank you.
(line 433)
- The authors should include in the discussion the limitation of their study such as the demonstration of butyrate treated NPICCs in reversing diabetes post-transplantation compared to untreated NPICCs.
Response 15: We agree with the reviewer that the limitations of this study should be mentioned in the discussion: We added this part at the end of the discussion.
There are some limitations of our study. We have not determined the molecular mechanisms
how inhibition of HDAC promote beta cell differentiation and maturation. Butyrate effects were only evaluated in vitro and not in vivo. Further studies are needed to explore whether treatment with butyrate or HDACi provide NPICCs with improved capacity to reverse diabetes after transplantation (lines 466-471)
Reviewer 2 Report
GENERAL COMMENTS:
Zhang and Coll. address the effects of butyrate on beta cell differentiation and maturation using Neonatal porcine islets-like clusters. They demonstrated that the effect of butyrate may be predominantly mediated through its HDAC inhibitory activity. Their data are interesting and suggestive that butyrate has beneficial contribution to increase functional beta cell mass in Neonatal porcine islets-like clusters.
Comments
- Any in vivo data? Butyrate group showed higher amount of beta cell mass. Did the authors transplant piglets to diabetic mice based on beta cell number or islet number? Similar function in vivo?
- Piglets treated by Butyrate have higher amount of matured beta cells but maybe lower number of progenitor cells. Can piglets treated by Butyrate survive longer in vivo?
Minor comments
- Figure 1 missing H, I
Author Response
We thank the reviewers for taking the time to review our manuscript.
Response to Reviewer 2 Comments
Zhang and Coll. address the effects of butyrate on beta cell differentiation and maturation using Neonatal porcine islets-like clusters. They demonstrated that the effect of butyrate may be predominantly mediated through its HDAC inhibitory activity. Their data are interesting and suggestive that butyrate has beneficial contribution to increase functional beta cell mass in Neonatal porcine islets-like clusters.
Comments
- Any in vivo data? Butyrate group showed higher amount of beta cell mass. Did the authors transplant piglets to diabetic mice based on beta cell number or islet number? Similar function in vivo?
Response 1: We are thankful for the reviewer’s suggestion. Indeed, it will be very interesting to analyse reversal of diabetes after transplantation of butyrate treated NPICCs in diabetic mice. Thus far, we have not performed these experiments. Because of this limitation we have now included a section to discuss this point.
There are some limitations of our study. We have not determined the molecular mechanisms
how HDACi promote beta cell differentiation and maturation. Treatment with butyrate was only evaluated in vitro and not in vivo. Further studies are needed to explore whether butyrate treatment provides NPICCs with improved capacity to reverse diabetes after transplantation (lines 466-471)
- Piglets treated by Butyrate have higher amount of matured beta cells but maybe lower number of progenitor cells. Can piglets treated by Butyrate survive longer in vivo?
Response 2: We agree that measurement of the survival rate of butyrate treated NPICCs would be very interesting. As mentioned above these experiments were not performed thus far.
Minor comments
- Figure 1 missing H, I
Response 3: H and I are now added to Figure 1 (page 7)